# The Challenges of Hospitals’ Planning & Control Systems: The Path toward Public Value Management

**DOI:** 10.3390/ijerph18052732

**Published:** 2021-03-08

**Authors:** Sabina Nuti, Guido Noto, Tommaso Grillo Ruggieri, Milena Vainieri

**Affiliations:** 1Management and Health laboratory, Institute of Management—Department Embeds, Sant’Anna School of Advanced Studies, 56127 Pisa, Italy; sabina.nuti@santannapisa.it; 2Department of Economics, University of Messina, 98122 Messina, Italy; gnoto@unime.it; 3Progettazione e Sviluppo del Sistema Informativo, Gestione dei DWH e dei Sistemi Decisionali, 16121 Genoa, Italy; tommaso.grilloruggieri@regione.liguria.it

**Keywords:** planning and control, healthcare, hospital, public value management, performance management

## Abstract

In the last decades, public management has been subjected to a shift from the New Public Management (NPM) paradigm to the Public Value Management (PVM) one. Thus, management practices such as Planning and Control (P&C) systems have been called to evolve accordingly. The health care sector has not escaped this process. This paper focuses on the evolution of hospitals’ P&C systems to support the paradigm shift from the NPM paradigm to the PVM one. In particular, the paper aims at exploring whether hospitals’ P&C systems in Italy evolved, or are evolving, consistently with PVM and what are the expected benefits related to such a paradigm switch. To address the research aim, the paper is based on a review of scientific and grey literature and the case study of the diabetic-foot pathway in an Italian Regional Healthcare System. The results of this study show that the current P&C systems in Italian hospitals are not yet designed to support the shift toward the PVM approach and are still mainly focused on financial aspects and intra-organizational dynamics. Combining traditional P&Cs with performance measures assessing the system’s outcomes may support hospitals in aligning their goals with the health system they are operating within and, therefore, P&C systems may represent an important driving force toward change. Such results provide suggestions for both practitioners and academics on how to adapt P&C systems to better support the implementation of current strategies of the public sector.

## 1. Introduction

Over the last thirty years, NPM reforms have drastically changed the role and the way public sector institutions are managed [1]. These reforms were aimed at addressing the shortcomings of the traditional paradigm of Public Administration, based on Weber’s model of ideal bureaucracy [1].

To do that, governments in the West have introduced concepts and tools derived from the private sector—e.g., disaggregation of administrations into corporatized units around products/services, competition, linking resources allocation to measured performance, and managerial responsibilities throughout the organizations [1].

Initially, NPM’s main focus was on the intra-organizational level. This phase is also known as “managerialism” or “managing for results” [2,3].

Managerialism was aimed at orienting public sector organization’s structure, financial management, staffing, and rewards toward the achievement of better outputs and productivity. It was modeled on the multidivisional form of private-sector corporations of the 1980s, i.e., breaking organizations down into various business units which were then controlled by setting and monitoring performance results [3]. This led to the adoption of the first P&C systems which, according to Amigoni [4] may be defined as those set of managerial tools oriented at setting goals, assessing performance, and taking corrective actions. The role of P&C tools in this reform phase was to support top managers to attribute individual responsibilities throughout their organization to be able to evaluate each intra-organizational unit performance [2,3]. 

The healthcare sector has been involved in this reform process in almost every country [5,6]. For instance, in the USA, UK, and New Zealand (see respectively: “Tax Equity and Fiscal Responsibility Act”, “Financial Management Initiative”, and “Financial Management Reforms”) a number of legislative packages or government initiatives were implemented during the deregulatory wave in the 1980s [5]. In Italy, at the beginning of the 1990s (see government decrees 502/1992 and 517/1993), an NPM oriented reform was also implemented to foster managerial autonomy of organizations within the Italian national healthcare system through devolution of powers and the introduction of management control tools [7]. 

Although NPM addressed many of the pitfalls of the bureaucratic model, it strengthened a silo structure that fostered competition among organizational units and institutions working in the same system, especially in terms of financial resources allocation [3,8]. In fact, the first response of NPM to the inefficiencies in the traditional public management approach was to fragment monopolistic service structures and to develop incentives and tools to promote productivity within every single department [6,9].

Due to the devolution processes, healthcare systems started to be structured according to a multi-level governance setting where various government levels and a multiplicity of providers operates. This enhanced institutional fragmentation that limited the decision-makers’ ability to improve public service provision [8]. Public organizations thus ran into issues such as the tension between different policies and agencies; duplication and contradiction of action programs, and fragmentation of service provision to citizens [3].

Many changes require a new approach to the performance management system in health care [10] coming from both external factors and the evolution of organizing care.

In fact, in the last decades, healthcare organizations have been also exposed to an increased complexity deriving from changes in epidemiological conditions—such as the aging population—which are determining new care needs related to the increasing emergence of chronic diseases. These require the implementation of specific clinical pathways in which a multiplicity of providers (e.g., general practitioners, hospitals, etc.) are called to collaborate [11,12].

Outcomes achieved along such pathways are thus related to this multi-provider, multi-disciplinary, and multi-professional service chain. These cannot be measured and evaluated by considering individual health services delivered by single providers but should embrace a systemic—or population-based—perspective [10,12,13]. In particular, “a population perspective encompasses whole populations (i.e., defined by geography, insurance coverage or attribution) and not only those with specific illnesses or needs” [10]. In healthcare, outcomes are considered to be a key determinant of the value created. Value in health can be therefore defined as multifaceted and multidimensional population outcomes achieved with the available resources—where population outcome refers to the ability to deliver services to those who could benefit most from it [13]. That means measuring what is provided but inappropriate as well as quantifying also the unmet needs.

To face these challenges, today, many countries are trying to strengthen the relationships between settings, organizations, and professionals, irrespectively of the healthcare system model adopted (e.g., Beveridge, Bismarck, or the private insurance-based one). The underlying idea is to make health care more accessible, usable, effective, and efficient—i.e., to provide more value not just to the patients cared for but to the whole population with the available resources [12,13].

This all entails a paradigm shift in the health care system management. New strategies have thus been implemented to foster integrated care, i.e., a complex concept which involves coordination, integration, and continuity of care to create value [11,14].

This paradigm shift is consistent with the global trend characterizing public management all over the world, i.e., PVM [15]. In fact, since the late 1990s, many countries have been striving to deal effectively with institutional fragmentation and to enhance coordination between different stakeholders and governance levels to improve public sector performance [3,8]. PVM has been defined by Stoker (p. 56) as the new paradigm for the public sector; its strength lies in the “redefinitions of how to meet the challenges of efficiency, accountability, and equity and in its ability to point to a motivational force that does not rely on rules or incentives to drive public service reform” [16]. The underlying concept of the value paradigm is thus the adoption of a systemic approach that takes into consideration the interests of several stakeholders in the healthcare system and embraces a population-based perspective. This allows one to overcome the exclusive focus on the financial performance and the productivity of every single unit/organization considered as a separate entity.

To the author’s knowledge, few studies focused on the contribution that P&C systems may provide in supporting a shift from managerialism to PVM. P&C systems are key tools to focus decision-makers on outcomes achievement rather than exclusively on the inputs and outputs already considered in traditional systems [10,17,18]. As such this paper suggests that new management accounting and control practices should go beyond efficiency and effectiveness measures to foster the renewed public sector strategy, namely: public value creation and delivery.

Based on these premises, the paper is aimed at highlighting the challenges and the opportunities of hospitals’ P&C systems to embrace the public value perspective overcoming the limits deriving from institutional fragmentation. In particular, the article aims to answer the following research questions:

RQ1: To what degree hospitals’ P&C systems have evolved to integrate new measures according to the paradigm shift from the first NPM phase to PVM?

RQ2: Which advantages may derive from the integration between traditional P&C and outcome-based measures at the system (or population) level?

To answer the research questions, this paper focuses on the role of P&C systems in hospital organizations in Italy. Analyzing data and real experiences, the study outlines and discusses the challenges and limits of the current P&C systems, paving the way toward potential further development.

The paper structured as follows: the next section explores the theory beyond P&C in healthcare; the third section outlines the material and methods used; the fourth section introduces the Italian context presenting original data related to hospitals P&C systems and discusses a case study related to the challenge of measuring performance in specific care pathways; the lasts sections provide with some conclusive considerations and practical implications.

## 2. Theoretical Background

P&C tools could be defined as those mechanisms used to set goals, assess performance, and take corrective action [4]. These were introduced into the public sector to support performance management—i.e., the process by which an organization manages its performance in line with its institutional mission, strategy, and desired purposes. Bouckaert & Halligan stated that the general purpose of these tools is to “allocate responsibility for the performance of a system and being accountable for its results” (p. 2) [9].

The advantages of performance management applied to the public sector are particularly important in considering the accountability requirement of linking activities and resources and the performance achieved by the administration (see among others [19,20]).

According to an instrumental perspective [18], the performance domains usually assessed by P&C systems in health [21,22] are threefold: input, output, and outcome.

Traditional P&C systems focus on input (e.g., costs) and outputs (e.g., the volume of treatments delivered) and have been mainly used on an annual basis to monitor the use of resources and to attribute organizational responsibilities [23,24].

Among traditional P&C systems, the budget is the most frequently adopted by public healthcare organizations [23]. Budgets, in their managerial function, help to identify the contributions and responsibilities of each organizational unit with regard to the overall economic performance.

However, budget systems have several limits that have been highlighted by both literature and practices (see among others [19,25,26,27,28,29]. In particular, budget systems:are often weakly linked to strategy and strategic plans [25];usually focus exclusively on financial measures [26];tend to cause dysfunctional behaviors [25,27,28,29] such as “myopia”, e.g., when managers focus on short-term targets at the expense of longer-term objectives;focus on sub-optimal performance—e.g., they try to maximize an organizational units’ performance neglecting its impact on the outcomes of the whole system [25,27].

As a result, the use of traditional budget systems have not supported decision-makers to properly assess policy impacts with reference to (a) the trade-offs existing between the multiple governance levels; (b) the multiple performance dimensions that characterize the public healthcare sector [12,19,20,23] leading also to some unintended consequences [30].

Traditional P&C systems are thus not suitable to support hospitals in the implementation of integrated care strategies previously mentioned. In fact, even though the objective of financial sustainability is particularly relevant for these organizations, P&C tools also need to consider the other dimensions of value creation [12,20,31] and to assess performance at the system level, i.e., the outcomes related to the multiple governance levels involved in the health system [12]. In fact, hospital traditional P&C systems often tend to monitor what is provided by the hospital in terms of treatments, interventions, rarely of quality and outcome at the individual level, hardly ever quality and outcome at the population level.

The above limits of budgeting systems derive from the need to design a performance assessment system consistent with the first NPM phase (i.e., managerialism). To overcome these limits, recently, some health organizations’ budgeting and P&C systems started to also set objectives considering outcomes at the single-institution level [32]—e.g., 30-day mortality after ischemic stroke, 30-day re-admission, etc. However, at the hospital levels, few P&C systems consider the achievement of outcomes at the system level, i.e., population-based outcomes (e.g., reducing avoidable hospitalizations), which better represents the value creation process. Thus, although the new PVM paradigm was introduced for almost two decades, there is still relatively little use of P&C tools reporting also a performance at the system level [17].

## 3. Material and Method

The research has been designed comprehending two main steps directly related to the research questions developed and presented in the introduction of the work. These comprehend
(i)a review of the extant literature on the topic of P&C of hospitals,(ii)the development of the state of the art of Italian hospitals P&C systems,(iii)the development of a case study related to a complex clinical care-path.

While the literature review is aimed at providing a state of the art of the evolution of P&C in Italian hospitals; the case study focuses on the integration between traditional P&C and outcome-based measures.

The literature review took into account both scientific papers and grey literature on the evolution of hospitals’ P&C systems. The exploratory review on the Italian hospital P&C systems was carried out through EBSCO database considering as keywords “budget” or “P&C” or “performance evaluation system” or “performance measurement systems” or “performance management systems” limiting the results to the Italian hospital setting. As a result, we obtained 49 articles to consider. Through an in-depth screening, 11 articles were selected and considered in the analysis. In the second section of the results, this evidence was then integrated by the information collected from grey literature with a special focus on the Italian reports periodically disclosed by the Italian research centers working in healthcare management.

Based on such a review, a state-of-the-art of Italian hospitals’ P&C practices has been developed to assess to what degree these evolved or are evolving consistently with the PVM paradigm.

For what concerns the development of the case study, we choose to focus on a longitudinal case of a complex care pathway whose activity comprehends the involvement of the hospital setting as well as primary and ambulatory care. This is the diabetic foot pathway [11]. We examined the case of Tuscany with a specific focus on the University Hospital of Pisa. In particular, the case study discussed the potential benefits related to the integration of traditional and system outcome-based P&C systems. Based on previous research [11], the case analyzes documents and primary data at both the regional and organizational levels.

## 4. Results

### 4.1. A Review of the Literature on Italian Hospitals’ P&C System

The Italian national health system (NHS) follows the Beveridge model. In particular, this system is financed through general taxation and provides universal coverage for comprehensive and essential health services. In this system, hospital care accounts for about 45% of NHS funds [33].

As a result of the NPM devolution reform in Italy, since the early 1990s, the Italian NHS has been organized on a regional basis (Italy is made up of 19 regions and 2 autonomous provinces). Based on the national health plan, each regional government defines its own health plans and strategies and allocates the budget to its health authorities which are in charge of delivering the related services. Regional governments provide health services through (i) Local Health Authorities (LHAs), geographically based organizations financed by capitation, which deliver public health, community health services, and primary care directly as well as secondary and specialist care through directly managed facilities, or by purchasing services from public hospital institutions or private accredited providers; (ii) autonomous/university public and private accredited hospitals focused on acute care and financed by service tariffs; and (iii) private not accredited providers financed by service tariffs.

In the late 1990s, organizations in the public healthcare sector in Italy moved from financial to accrual accounting and also introduced budgeting systems as the key P&C tool [34]. The adoption of these tools in hospitals was aimed to set strategic and managerial objectives to top managers and heads of department (HoD) and to attribute the related responsibilities [35,36]. Top managers were thus enabled to assess their contribution to the overall organization results in financial terms.

Anessi-Pessina and Cantù [34] outlined three main characteristics of the budget systems in the Italian health organizations: they are highly structured in hospitals compared to other settings of care; they often do not report a reference to external benchmarks or best practices; they require a long planning and control cycle.

The importance of introducing indicators representing professional reputations into the hospital budget system has been highlighted by several authors. Burns and Scapens [37] maintained that management accounting system, like the budget, needs to be institutionalized to effectively change the daily routines of those who have the decision powers. This means that health professionals in charge of managerial decisions should perceive that the budget system provides them with useful information for their activities and their role as managers. From this perspective, Macinati [38] reported that “clinicians use of the accounting terminology was limited, and financial information was not used in decision-making”, also because the goal commitment depends upon the participation in the budget process [39].

As regards the type of goals, Ancarani et al. [40] found that Italian hospital productivity (of both medical and surgical units) was positively related to the professionals’ intention to pursue the prestige; conversely, net returns (revenues net of direct and indirect costs) were negatively related to efficiency. Similarly, Bravi et al. [41] found that the relevance of quality of care was one of the most important factors that healthcare professionals identified for high performant Italian hospital networks. In this sense, to engage effectively HoDs with the organizational values (which is a predictor of the implementation of the budget [42]) it is important to include into the budget system goals which are close to the healthcare professionals’ values. Indeed, Bosa [43] pointed out that management accounting systems can be successfully implemented when there is an alignment between the management imperative and the ethical framework in which doctors practice their profession. From this standpoint, the inclusion of outcome indicators, as well as practice and appropriateness indicators were chosen by professionals into the budget system, may ease the process of goal alignment.

### 4.2. The State of the Art of Outcome Measure Integration within the Italian Hospitals’ P&C System

The integration of quality of care and outcome measures on budgets requires the availability of such information and its evaluation according to predefined criteria.

At the Italian level, the 2016 Report by the Expert Group on Health Systems Performance Assessment [44] selected three experiences defining, computing, and evaluating quality indicators, namely: (i) the Essential Levels of Healthcare (ELC) Grid, implemented for all Italian regions, ensuring evaluation of the homogenous provision of the essential services to be granted for all Italian citizens; (ii) the National Outcomes Evaluation Programme (NOEP), implemented for all Italian regions, that investigates the heterogeneity of access to health care focusing on health care interventions for which evidence of effectiveness is available; (iii) the Inter-Regional Performance Evaluation System (IRPES), implemented for a network of Italian regions voluntarily, that monitors multiple dimensions of healthcare services.

The ELC Grid was introduced by the Ministry of Health as an assessment tool in 2010. Its 32 indicators are measured at the regional level and are calculated with reference to the resident population, with the exception of hospital indicators of efficiency and appropriateness [44].

The NOEP was introduced in Italy in 2010, it was developed by the National Agency for Healthcare Services on behalf of the Ministry of Health [34]. This system aims at monitoring health outcomes nationwide and supporting clinical and organizational audit programs. NOEP accounts for more than 100 indicators calculated based on administrative data and classified in clinical areas. These indicators show the results of the different organizations and are available at the hospital and/or the province level.

The IRPES started in 2008, it entails multi-dimensionality, evidence-based data collection, shared design, systematic benchmarking of results, transparent disclosure, and timeliness [12,44]. The IRPES includes more than 100 indicators that measure the multidimensional performance of each healthcare organization and take into account: population’s health status; capacity to pursue regional strategies; clinical performance; efficiency and financial performance; patient satisfaction; and staff satisfaction.

IRPES and NOEP, although developed according to different criteria and purposes [45], represent the two main macro systems that publicly disclose the performance obtained by the Italian hospitals. As previously mentioned, both systems focus on outcome results and multiple performance dimensions that go beyond financial measures and volumes of services provided.

Although these two systems provide performance information in benchmarking for all Italian hospitals (NOEP) or a selection of Regions (IRPES), the introduction of such systems has not led to a full integration and use of these indicators into the hospital budgets [46].

10 years since the NPM reform in Italy, about 80% of health authorities introduced the budget system [47]. Management accounting systems registered high delay in both the introduction and the process (the final goals were often set in April/May), and often the budget implementation was only executed at a formal level. Selected Italian case studies reported that the importance of the budget system increased after several health authorities merged in 2015; these budgets started including both financial and non-financial measures [48]. However, as highlighted by surveys of 39 controllers conducted in 2016, the weight of financial indicators was exacerbated by the first years of the financial crisis (whose effects lasted until the pandemic); interviewed declared that half of the objectives of their next budget cycles were related to cost containment [49].

To understand how much these systems or other types of goals are acknowledged by health professionals, we focused on the organizational climate reports based on surveys conducted in seven Italian Regions belonging to the IRPES network (Toscana, Lombardia, Veneto, Umbria, Marche, Puglia, and Emilia Romagna) [50,51,52,53,54,55,56,57,58]. In these reports, there is a thematic section investigating the perception of the HoDs upon budget systems. In particular, they were asked to remember which type of indicators were included in their budget system and also pointing out from which of the regional and national systems previously mentioned (IRPES and NOEP) they were coming. In Table 1, we reported a picture of the information included in the budget systems of the twenty Italian hospitals involved. These surveys were carried out in 2016–2019 and were administered to all of the HoDs. The response rate was on average 50% with a total amount of 1068 responses collected.

The result of the surveys shows that the large majority of hospital departments budget includes the number of treatments in terms of “cost information” for 81%; information about “revenues” (mainly services financed according to DRG tariffs) for 67%. 64% of HoDs stated that the budget comprehends information about “personnel”. While the percentages of HoDs claiming that their budget systems comprehend information about users’ experience, the NOEP, and the IRPES systems in the 20 hospitals analyzed are 46%, 38%, and 35%, respectively (see Table 1).

These findings suggest that hospital budgets are still mainly linked to financial and input dimensions (revenues, costs, and personnel). Interestingly, the majority of the hospitals analyzed are paying attention to the clinical risk dimension which indicates that hospitals started including indicators related to quality and outcome at the single institution level.

This empirical evidence suggested that, although there are national and regional performance evaluation systems (namely NOEP and IRPES), the Italian hospital budget systems are still mainly oriented toward the NPM managerialism approach—or the HoDs are not aware that their budgets include indicators from NOEP and IRPES. In any case, they are not focusing yet on the wider system perspective. In fact, Italian hospitals’ P&C systems are not considering indicators and objectives related to system-level outcomes and other quality or equity measures in a systematic manner. As a result, single hospitals or departments often pursue objectives based on volumes of services and financial measures, whose results do not necessarily contribute to the overall value creation.

### 4.3. Integrating Hospital’s P&C Systems with Population-Based Outcome Measures: The Experience of Diabetic Foot Pathway in Tuscany

In healthcare pathways (e.g., chronic care paths) each phase, such as care services, is usually under the responsibility of a different provider (e.g., hospital, general practitioners, specialists, etc.) and the integration between acute hospital care and other care settings is crucial for the achievement of good outcomes. In this regard, the adoption of a value-based perspective and benchmarking practices may be used to highlight the results jointly achieved for the population health by the system of healthcare providers. Indeed, including in P&C systems input, output or outcome measures exclusively related to a specific phase and/or provider is likely to incentivize a silo inward-looking perspective. For those care pathways in which value is pursued through a joint and interrelated set of interventions and treatments, considering the population outcome measures driven by the multi-provider care continuum may support the alignment of all the professionals and providers towards the same strategic goals. This value-based perspective may also be useful to guide resource reallocation toward the activities relevant for the population value creation process rather than increasing every single provider/unit result.

This section presents the experience conducted in Tuscany Region on the diabetic foot care pathway as a case study showing how focusing on population outcome targets may support decision-makers in shifting the attention from organizational objectives toward health system ones. This chronic disease pathway perfectly embodies the complexity of healthcare multi-provider, multi-professional and multi-disciplinary care paths.

This experience started in 2012 when the outcome indicator of the hospitalization rate for diabetes-related amputations was discussed between the researchers of the IRPES and the Tuscan professionals [11]. In particular, after a constructive research approach, the Tuscany Region issued a protocol for the treatment of the patient with a diabetic foot. This indicator highlights the lack of coordination among the different settings of care and professionals involved managed to deliver preventative treatments which preserve the patient conditions without giving rise to acute phases. Usually, this kind of outcome is attributed exclusively to the health authorities responsible for the population health, namely the LHAs. Figure 1 provides details about the hospitalization rate; in 2012, the LHA of Pisa (a province in Tuscany) was the worst performer (residents in Pisa LHA accounted for a higher rate of diabetes-related amputations compared to the other LHAs). Its amputation rate was near twice the regional average and six times higher than the best performing LHA (with persistent similar results through the years). Around 80% of the Pisa amputations were provided by the Pisa University Hospital (Pisa UH). The institutional separation between the LHA in charge of delivering primary, home, and community care and the Pisa UH, in charge of delivering hospital care and financed through DRG tariffs, was recognized as a major reason for the complexity of the coordination between the hospital, community and primary care services in this area. Conversely, the best performer, namely Arezzo LHA, was mainly providing these services to its reference population internally (i.e., without recurring to other public or private hospitals) [11].

Hospitals are usually made responsible for the quality of treatment they deliver. This, together with the role of the leading institution in diabetic foot care at the regional level and the lack of emphasis over a system perspective in the UH was causing an underestimation of the impact of a volumes-driven perspective on the population. The focus of UH was indeed on delivering quality services to hospitalized patients without taking care of the previous phases of the pathway.

In this sense, the IRPES indicator through publicly benchmarking performance in terms of outcomes per population served as the first, necessary, step for driving improvement. It shed light on the need for improvement in outcomes of the diabetic foot multi-provider value chain. It also helped to shift the focus from input and outputs toward systems’ outcomes (i.e., population health).

The traditional P&C systems adopted at that time were not able to support the change in perspective needed, i.e., from output to value because it did not consider the rate per population but the percentage on the cases treated by the hospital. On the one hand, traditional internal P&C tools were further emphasizing an inward-looking perspective on the control and efficient use of the UH resources. On the other hand, pressing the Pisa LHA to reduce the high incidence of diabetes-related amputation in the Pisa-LHA were not solving this issue, since most of these surgical operations were delivered by the Pisa UH and not by the LHA facilities. As a result, the LHA could not effectively intervene in the UH delivery system.

In 2012 Tuscan professionals working in hospitals both from LHA and UH, started monitoring and using those outcome indicators to improve their performance and the Region decided to set a shared quantitative target on the reduction of the diabetes-related amputation rate to both LHA and UH. This objective was then declined to the heads of the department involved in the pathway.

Moreover, the study conducted in 2012 [11] highlighted also different practices about revascularizations. In particular, the best and worst performers used a different mix of treatments (revascularization or amputation) whose cost per population was almost the same. In particular, there was not a big gap in the overall budget per capita allocated by the best-practice (Arezzo LHA) compared to Pisa LHA. The real difference was related to the mix of services. Indeed, Arezzo LHA does not account for the overall lower cost per 100,000 residents but accounts for a cost mix mostly oriented toward preventative services, being able to achieve more value for patients with the same amount of resources as other LHAs with poorer outcomes [11]. In 2013 Tuscan health system, on the thrust of healthcare professionals who participated in the analysis, issued a regional protocol to integrate diabetic foot pathway (and diabetes management in general) between primary care and acute care.

To better understand the effects of this shared protocol we analyzed data between 2010 and 2017: four years before and four years after the issue of the regional protocol—we considered 2013 as a transitional year thus we included it in the first period of analysis.

The hospitalization rate for major amputation decreased at the regional level going from 53 (2010–2013) to 46 (2014–2017) per million inhabitants as well as for Pisa province that passed from 97 (2010–2013) to 64 (2014–2017) per million of inhabitants. The sharpest reduction of Pisa (around 35%) was also due to the involvement of the head of the center for the diabetic foot of Pisa who promoted the organizational benchmarking among Tuscan hospitals.

The two main cost items associated with the diabetic foot pathway are (i) severe amputations and (ii) revascularization (i.e., preventative treatments). These were estimated and benchmarked for every Tuscan LHA.

The measure adopted allowed the professionals involved in the multi-provider care pathways to overcome the organizational incentives linked to a volume-based and silos perspective. In fact, it supported and incentivized organizations and professionals to work for reallocating resources towards a service-mix that increases population value. In the case analyzed, the services-mix of amputations and revascularizations in the Pisa area sharply changed towards the preventative interventions (i.e., revascularizations) (see Table 2).

Table 2 shows that, although the cost per resident slightly reduced in Pisa against an increase at the regional level with also a different cost-mix. In particular, the cost per amputation decreased by 32% while the cost per amputation in the whole of Tuscany decreased by 8%. The result of this different resource allocation in the case of Pisa led not only to a reduction of the overall costs but also to higher gains in terms of population outcome (linked to a reduction of amputations).

According to the evidence provided by this case study, the results achieved in term of value creation is mainly attributable to the reputational mechanism [11,59] enabled during the 2012 research project and kept vivid from the integration of the IRPES information with the Pisa UH and LHA budgets. It is clear however that we cannot exclude that other external factors may have influenced in part the outcome achieved. This case demonstrates that the integration of hospitals’ P&C systems with PVM outcome measures may foster the overall health system performance.

## 5. Discussion

Healthcare systems are characterized by a set of features that tend to lead to a high level of institutional complexity—e.g., multi-level governance structures; a multiplicity of stakeholders and providers; asymmetric information between providers and users.

Despite the epidemiological changes characterizing modern societies, the hospital setting may still be considered as one of the main pillars of healthcare systems all over the world accounting for a share that ranges from 25 to 50% of the total health expenditure [60].

What emerges from the literature and the experiences here presented is that current hospitals’ P&C systems are evolving but they are not yet designed to cope with the emerging need of adopting a systemic and inter-institutional perspective to foster PVM.

According to the surveys’ results displayed in the third section, the main focus of hospital budgets in Italy are still cost, volumes and revenues. Information related to performance indicators on quality, outcomes coming from regional or national monitoring systems are ranked in the last positions. The spread presence of both cost and revenues in the budgets demonstrates that Italian hospitals and departments tend to use profit as a key performance indicator. A department’s net-income, calculated as revenues (which depend on the number of inpatient services multiplied by their specific DRG tariff) minus costs (human resources, equipment, pharmaceutical expenses, and a share of indirect and general costs), represents the financial result created by each department. As supported by Guthrie and English [2], in the public sector, profit is not a measure of performance, especially in a single payor system pursuing universal coverage financed through capitation like the Italian NHS. Although financial performance may be important for a hospital from the perspective of an individual institution, its related mechanisms (e.g., excessive focus on the DRG system) may undermine the overall financial sustainability of the broader health system and generate unintended consequences which compromise the creation of value for the reference community. In fact, in Beveridge systems financed based on capitation, hospitals revenues are paid by other health authorities and, therefore, represents costs at the overall system level. Moreover, focusing on revenues may encourage professionals to provide the service mix that maximizes revenues through the financially optimal mix of treatment volumes rather than considering the relevance of appropriateness and quality of care [61].

In this context, budgets and other P&C systems that are mainly tied to the financial dimension of performance are not suitable and may even compromise broader health systems’ strategies. Their application may encourage focusing on administrative data rather than clinical processes and, as a result, increasing the hospitals’ financial performance but depleting the system value creation capacity. This statement does not suggest the dismissing of the budget as a key management tool to orient decisions at the organizational level [62]. However, this should be adapted or integrated with other macro-level PM systems to answer the need for considering hospitals as part of a broader health system.

According to the case developed in the fourth section of this article, integrating hospital P&C systems with the system’s outcome measures and targets may represent a solution to foster collaborative governance and to support the creation of population value as defined by Gray et al. [13]. The integration between national or regional systems with organizational P&C may support not only introducing outcomes and quality indicators but also stimuli to overcome self-referral attitude throughout benchmarking.

Although some of the budgets of Italian hospitals already include indicators related to external referenced systems (such as NOEP and IRPES), this percentage should increase to better integrate traditional P&C systems with systems’ outcome measures. As such, a cultural change in the utilization of P&C by controllers and HoDs is required. According to the data collected from the organizational climate surveys data reported in Section 4.2, it emerges how this cultural change is ongoing. In fact, the percentage of hospitals that are now including in their budget information coming from the IRPES and NOEP system seems to increase (see median higher than mean in Table 1). Enhancing the relevance of outcome indicators may enable reputational mechanisms [59].

Further studies are needed to confirm the results emerging from the case study here developed. Moreover, future studies could focus on the key elements characterizing a successful integration of traditional and outcome-based measures and outline a clear process toward it.

## 6. Conclusions

According to the research here presented, it is possible to outline the following key results. First of all, the paradigm switch between NPM and PVM is still an ongoing process that requires additional effort at all the organizational and governance levels. Second, P&C systems may play an important role in supporting this shift and, therefore, in guiding public hospitals and health systems toward the creation of value for the reference community. Third, to do that, hospitals’ P&C systems should include population outcome measures and should consider benchmarking practices to evaluate the related performance.

Italian hospital P&C systems seem to include start including outcome-based indicators and indicators coming from upper-level systems to promote the comparisons. Albeit this evidence, there is wide variation across hospitals. The finding shows that current P&Cs in Italy are still mainly focused on financial aspects and intra-organizational dynamics. According to this result, hospitals should thus improve their effort to integrate new population outcome measures in their P&C system to support the creation of value for the population they serve. This integration mainly translates into a target setting and monitoring activity throughout healthcare organizations that use population outcome performance information according to the organizational level analyzed and its contribution to the whole healthcare system objectives.

Indeed, the case study of the diabetic foot in the Tuscan health system pointed out that when professionals (although those from the hospital setting) participate in the analysis of population data (such as the amputation rate) and also their determinants, they enable several actions that change their routines and also overcome institutional barriers. Moreover, reporting the cost per disease at the population level may reveal that even though the choices related to a different mix of treatments are not relevant from a financial point of view; they make the difference from the patient perspective as in the case of the diabetic foot pathway. In addition, reporting performance at the population level may also point out that behind similar strategies there can be different care path as it emerged from a recent study on innovative anti-cancer drugs [63].

The limitations of this research are mainly related to the exclusive focus on the Italian context both in the literature review and in the longitudinal case study. Therefore, further studies may be addressed by conducting a comparative analysis with other European and non-European countries. These studies may be aimed at analyzing the path from NPM to PVM characterizing different countries with different health systems to understand barriers and determinants of such process. Moreover, future research may explore to analyze in-depth the inclusion of system-level indicators into the hospital P&C systems rather than exclusively considering the results of each organizational unit seen as an independent entity. Indeed, the lack of a systemic perspective in the hospital settings may spur dysfunctional behaviors among organizations operating within the same health system. Last, this paper is mainly focused on a budget because it represents one of the most adopted P&C tools by the healthcare organization. This may however limit the generalization of some results when referring to other P&C mechanisms. Therefore, further researches may focus on other tools used to plan and control hospital performance.

## Figures and Tables

**Figure 1 ijerph-18-02732-f001:**
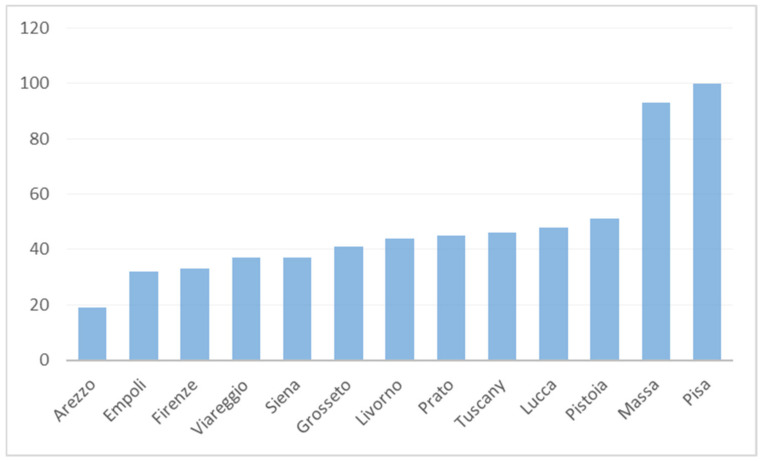
Diabetes-related amputation standardized rate per million residents in Tuscany LHAs (2012).

**Table 1 ijerph-18-02732-t001:** Organizational climate data on heads of department (HoD) use of the budget (percentages) ^1^.

Type of Information	Mean	Median	Min	Max
Costs	81	83	50	96
Clinical risk	68	67	38	92
Revenues	67	71	38	95
Personnel	64	69	22	92
Patients’ experience	46	45	3	81
NOEP indicators	38	39	9	79
IRPES indicators	35	38	5	85

^1^ Our elaboration on Management and Health Lab Regional report of health care organizational climate 2016–2019 [50,51,52,53,54,55,56,57,58].

**Table 2 ijerph-18-02732-t002:** Health expenditure by type of intervention ^1^.

Type of Information	Pisa		Tuscany	
	2010–2013	2014–2017	2010–2013	2014–2017
Revascularizations	€ 274,710 (84%)	€ 50,662 (16%)	€ 232,789 (88%)	€ 267,013 (12%)
Amputations	€ 274,710 (90%)	€ 34,553 (11%)	€ 31,030 (90%)	€ 28,410 (10%)
Total cost	€ 325,372 (100%)	€ 328,990 (100%)	€ 263,820 (100%)	€ 295,426 (100%)

^1^ Estimated LHA expenditure of hospitalizations for diabetes-related revascularizations and lower limb amputation per 100,000 residents in Pisa—Average of the four-year period between 2010–2013 and 2014–2017.

## Data Availability

The data presented in this study are available on request from the corresponding author. The data are not publicly available due to the agreement with the funding institutions.

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
