# Peer review of "The Challenges of Hospitals’ Planning & Control Systems: The Path toward Public Value Management"

_ijerph, 2021, doi:10.3390/ijerph18052732_

Round 1

Reviewer 1 Report

The article concerns an important and rarely undertaken topic, which is the efficiency of hospital management and the use of solutions known from the area of ​​management sciences for this purpose.
The main and essentially fundamental objection to the work is the unacceptable quality of the charts (Fig. 1, Fig. 2) presented in the paper. They are unreadable, with insufficient resolution. They need to be corrected.
The paper presents the practical application of two systems used in the hospital management model: New Public Management - based on the traditional managerial model (so-called managerialism) and a new concept: Public Value Management - using Planning and Control. This approach is correct, the reviewer finds it correct and scientifically sound.
In the opinion of the reviewer, the budgeting of costs was too clearly equated with the Planning and Control systems, because the P&C process itself may and should include numerous other tasks (the so-called controlling), such as personnel management planning, risk management, and medical process management. Thus, it is recommended to pay attention to the fact that budgeting itself is only one of the tools, not the only one in the P&C model. This is all the more important because for the operation of hospitals - as the authors point out - financial management is of secondary importance to process management.

Author Response

Reviewer 1

Thank you for your appreciation to our work and for the useful suggestions provided. We appreciate your inputs and modified the text accordingly. In particular:

R1: The motivation, justification and contribution must be more emphases in introduction section and in the abstract;

A: We modified both abstracts and the last part of our introduction to make the motivation and research objective clearer

R1: Material and methods section should be extended. Without clearly presents research methodology the paper has lower impact. I propose to show flowchart of all research.

A: Material and method section was improved. In particular, we tried to better explained the research design. We did not create a flowchart in order to avoid an excessive use of figures and tables, but we take advantage of the use bullet points

R1: Figures 1 - 2 are low quality. Please improve it

A: We replaced figure 1 with a clearer graph and figure 2 with a table

R1: In the conclusion section please extend the future research direction part

A: As you will notice, we improved the last part of the conclusion outlining some future researches

Reviewer 2 Report

The challenges of hospitals’ Planning & Control Systems: the 2
a path toward Public Value Management. This is a very interesting manuscript, rather complex, difficult to read and to follow. Besides, is very focused on the Italian hospital and health care management systems with a focus on one region.

I suggest the following modifications to make it more accessible to the average physician that may be at some level of management somewhere in the world.
1)    Divide the publication into short sections. Concise writing without long sentences. You are writing in English and not translating from Italian to English. Short, precise, concise.
2)    Maybe a diagram with a flow chart of the two comparative managerial systems would help to follow the work and understand the results
3)    The model of the diabetic foot is an excellent choice and deserves a well separate heading. It is almost a paper on itself. Maybe that should be the focus of the publication as it is a model that health care workers and administrators will understand.
4)    Who is your target audience?. It is not clear. As it seems very limited to the Italian Public Management  Sector that from this document is very complex. Do you have universal government-supported health care? Or more a private hospital model?. It is not clear cut in the manuscript. Or use multiple insurers systems?. 
5)    The Public Value Management approach makes sense but can you provide an example of an organization that actually uses it and that have used a model like yours to compare,
6) many very interesting connotations throughout the paper are lost in over the extensive document. Maybe the could be summarized and highlighted in the conclusions. Example professional personal value, vs financial value for the professional or the hospital vs value for the patient, or the national public health ??
7)    There is no account for preventive measurements and in-home care. The diabetic foot is an excellent model. Keep patient care at home. That will translate into less hospitalization etc.

Author Response

Thank you for your useful suggestions. We believe they guide us toward a better version of the manuscript. In particular:

R2: Divide the publication into short sections. Concise writing without long sentences. You are writing in English and not translating from Italian to English. Short, precise, concise.

A: We tried to improve our writing

R2: Maybe a diagram with a flow chart of the two comparative managerial systems would help to follow the work and understand the results

A: We did not create a flowchart in order to avoid an excessive use of figures and tables, but tried to address your point in the writing

R2: The model of the diabetic foot is an excellent choice and deserves a well separate heading. It is almost a paper on itself. Maybe that should be the focus of the publication as it is a model that health care workers and administrators will understand.

A: We appreciate your suggestion, but we believe that also the review of scientific and grey literature provide interesting insights. However, to make things clearer, we divided the results intro three sections (and not two as in the previous version).

R2: Who is your target audience?. It is not clear. As it seems very limited to the Italian Public Management  Sector that from this document is very complex. Do you have universal government-supported health care? Or more a private hospital model?. It is not clear cut in the manuscript. Or use multiple insurers systems?. A: In the first part of our results we tried to better explained the Italian NHS, its universal coverage and its financing system. Moreover, we acknowledged in the limitation our exclusive focus in the Italian context.

R2: The Public Value Management approach makes sense but can you provide an example of an organization that actually uses it and that have used a model like yours to compare

A: This is a very interesting insight; however it is difficult to label organization on the adoption of a unique paradigm. In most of case, in reality, healthcare organizations are in a continuous evolution process which use elements of various public management paradigm (Weberian bureaucracy, NPM, PVM) – see for instance Iacovino et al., 2015. Therefore, we do not want to run the risk to simplify this complex issue to the reader. We hope you could agree with that.

R2: many very interesting connotations throughout the paper are lost in over the extensive document. Maybe the could be summarized and highlighted in the conclusions. Example professional personal value, vs financial value for the professional or the hospital vs value for the patient, or the national public health A: We tried to summarize and highlight the key result in a first new paragraph in the conclusion section.

R2: There is no account for preventive measurements and in-home care. The diabetic foot is an excellent model. Keep patient care at home. That will translate into less hospitalization etc.

A: We exclusively accounts for the activity expenditure (revascularization vs amputation).

Reviewer 3 Report

Management practices such as Planning and Control (P&C) systems have been called to evolve accordingly. Therefore, it is very important research subject. Besides, the health care sector is especially related to management (insufficiency of resources). This work shows on the evolution of hospitals’ P&C systems to support the paradigm shift. The authors explore the current status of hospitals’ P&C systems in Italy so as to advance knowledge in the related field. The work is based on the documents’ analysis and on the case study of the diabetic-foot pathway in an Italian Regional Healthcare System. This research presents two main streams of literature which are related to the organizational paradigm of the public sector – with particular emphasis on the health sector – and the management control literature. The study provides suggestions for both practitioners and academics on the innovations to be introduced in P&C systems to better support the implementation of current strategies of the public sector. The paper is well written scientific work. Before accepting some shortcomings must be improved. The list of comments is as follows:
1. The motivation, justification and contribution must be more emphases in introduction section and in the abstract;
2. Material and methods section should be extended. Without clearly presents research methodology the paper has lower impact. I propose to show flowchart of all research.
3. Figures 1 - 2 are low quality. Please improve it
4. In the conclusion section please extend the future research direction part

Author Response

Thank you for having appreciated our work and for having provided us with useful comments and suggestions. We modified the text accordingly. In particular:

R3: The main and essentially fundamental objection to the work is the unacceptable quality of the charts (Fig. 1, Fig. 2) presented in the paper. They are unreadable, with insufficient resolution. They need to be corrected.

A: we changed them with better resolution ones

R3: it is recommended to pay attention to the fact that budgeting itself is only one of the tools, not the only one in the P&C model. This is all the more important because for the operation of hospitals - as the authors point out - financial management is of secondary importance to process management.

A: This is a very important issue and thank you for having pointed it out. We acknowledged in the last section of our work this limitation.

Round 2

Reviewer 2 Report

Major improvements thanks for answering the questions and for doing the revisions

Reviewer 3 Report

The paper can be acceppted after English language corrections.